# ACTION CONCEPT GROUNDING NETWORK FOR SEMANTICALLY-CONSISTENT VIDEO GENERATION

## ABSTRACT

Recent works in self-supervised video prediction have mainly focused on passive forecasting and low-level action-conditional prediction, which sidesteps the problem of semantic learning. We introduce the task of semantic action-conditional video prediction, which can be regarded as an inverse problem of action recognition. The challenge of this new task primarily lies in how to effectively inform the model of semantic action information. To bridge vision and language, we utilize the idea of capsule and propose a novel video prediction model **A**ction **C**oncept **G**rounding **N**etwork (ACGN). Our method is evaluated on two newly designed synthetic datasets, CLEVR-Building-Blocks and Sapien-Kitchen, and experiments show that given different action labels, our ACGN can correctly condition on instructions and generate corresponding future frames without need of bounding boxes. We further demonstrate our trained model can make out-of-distribution predictions for concurrent actions, be quickly adapted to new object categories and exploit its learnt features for object detection. Additional visualizations can be found at `https://iclr-acgn.github.io/ACGN/`.

## 1 INTRODUCTION

Recently, video prediction and generation has drawn a lot of attention due to its ability to capture meaningful representations learned though self-supervision (Wang et al. (2018b); Yu et al. (2019)). Although modern video prediction methods have made significant progress in improving predictive accuracy, most of their applications are limited in the scenarios of passive forecasting (Villegas et al. (2017); Wang et al. (2018a); Byeon et al. (2018)), meaning models can only passively observe a short period of dynamics and accordingly make a short-term extrapolation. Such settings neglect the fact that the observer can also become an active participant in the environment.

To model interactions between agent and environment, several low-level action-conditional video prediction models have been proposed in the community (Oh et al. (2015); Mathieu et al. (2015); Babaeizadeh et al. (2017); Ebert et al. (2017)). In this paper, we go one step further by introducing the task of *semantic action-conditional video prediction*. Instead of using low-level single-entity actions such as action vectors of robot arms as done in prior works (Finn et al. (2016); Kurutach et al. (2018)), our task provides semantic descriptions of actions, e.g. *"Open the door"*, and asks the model to imagine *"What if I open the door"* in the form of future frames. This task requires the model to recognize the object identity, assign correct affordances to objects and envision the long-term expectation by planning a reasonable trajectory toward the target, which resembles how humans might imagine conditional futures. The ability to predict correct and semantically consistent future perceptual information is indicative of conceptual grounding of actions, in a manner similar to object grounding in image based detection and generation tasks.

The challenge of action-conditional video prediction primarily lies in how to inform the model of semantic action information. Existing low-level counterparts usually achieve this by employing a naive concatenation (Finn et al. (2016); Babaeizadeh et al. (2017)) with action vector of each timestep. While this implementation enables model to move the desired objects, it fails to produce consistent long-term predictions in the multi-entity settings due to its limited inductive bias. To distinguish instances in the image, other related works heavily rely on pre-trained object detectors or ground-truth bounding boxes (Anonymous (2021); Ji et al. (2020)). However, we argue that utilizing a pre-trained detector actually simplifies the task since such a detector already solves the major difficulty by mapping high-dimension inputs to low-dimension groundings. Furthermore, bounding boxes cannot

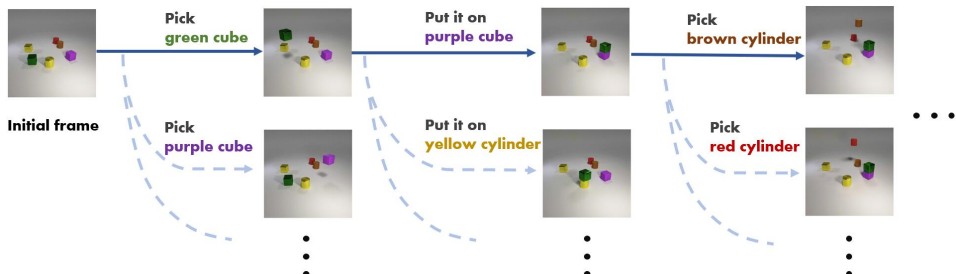

Figure 1: Semantic action-conditional video prediction. An agent is asked to predict what will happen in the form of future frames if it takes a series of semantic actions after observing the scene. Each column depicts alternative futures conditioned on the first outcome in its previous column.

effectively describe complex visual changes including rotations and occlusions. Thus, a more flexible way of representing objects and actions is required.

We present a new video prediction model, ACGN, short for Action Concept Grounding Network. ACGN leverages the idea of attention-based capsule networks (Zhang et al. (2018a)) to bridge semantic actions and video frame generation. The compositional nature of actions can be efficiently represented by the structure of capsule network in which each group of capsules encodes the spatial representation of specific entity or action. The contributions of this work are summarized as follows:

1. We introduce a new task, semantic action-conditional video prediction as illustrated in Fig 1, which can be viewed as an inverse problem of action recognition, and create two new video datasets, CLEVR-Building-blocks and Sapien-Kitchen, for evaluation.
2. We propose a novel video prediction model, Action Concept Grounding Network, in which routing between capsules is directly controlled by action labels. ACGN can successfully depict the long-term counterfactual evolution without need of bounding boxes.
3. We demonstrate that ACGN is capable of making out-of-distribution generation for concurrent actions. We further show that our trained model can be fine-tuned for new categories of objects with a very small number of samples and exploit its learnt features for detection.

## 2 RELATED WORK

**Passive video prediction**: ConvLSTM (Shi et al. (2015)) was the first deep learning model that employed a hybrid of convolutional and recurrent units for video prediction, enabling it to learn spatial and temporal relationship simultaneously, which was soon followed by studies looking at a similar problem Kalchbrenner et al. (2017); Mathieu et al. (2015). Following ConvLSTM, PredRNN (Wang et al. (2017)) designed a novel spatiotemporal LSTM, which allowed memory to flow both vertically and horizontally. The same group further improved predictive results by rearranging spatial and temporal memory in a cascaded mechanism in PredRNN++ (Wang et al. (2018a)) and by introducing 3D convolutions in E3D-LSTM (Wang et al. (2018b)). The latest SOTA, CrevNet (Yu et al. (2019)), utilized the invertible architecture to reduce the memory and computation consumption significantly while preserving all information from input. All above models require multiple frames to warm up and can only make relatively short-term predictions since real-world videos are volatile. Models usually don't have sufficient information to predict the long-term future due to partial observation, egomotion and randomness. Also, this setting prevents models from interacting with environment.

**Action-conditional video prediction**: The low-level action-conditional video prediction task, on the other hand, provides an action vector at each timestep as additional input to guide the prediction (Oh et al. (2015); Chiappa et al. (2017)). CDNA (Finn et al. (2016)) is representative of such an action-conditional video prediction model. In CDNA, the states and action vectors of the robotic manipulator are first spatially tiled and integrated into the model through concatenation. SVG (Denton & Fergus (2018)) was initially proposed for stochastic video generation but later was extended to action-conditional version in (Babaeizadeh et al. (2017); Villegas et al. (2019); Chiappa et al. (2017)). It is worth noticing that SVG also used concatenation to incorporate action information. Such implementations are prevalent in low-level action-conditional video prediction because the action vector only encodes the spatial information of a single entity, usually a robotic manipulator (Finn et al. (2016)) or human being. A common failure case for such models is the presence of multiple affordable entities (Kim et al. (2019)), a scenario that our task definition and datasets focus on.

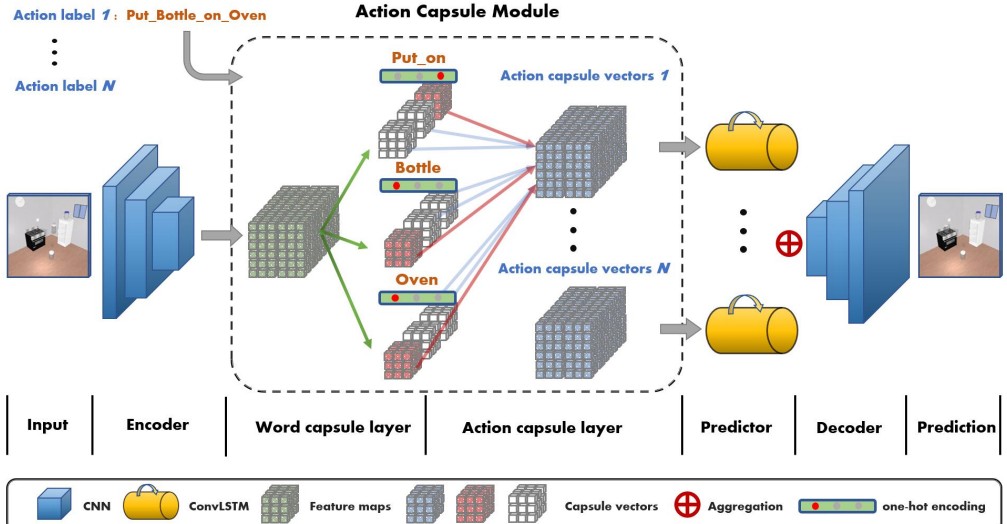

Figure 2: The pipeline of ACGN in which the computation of action capsule module is elaborated (Better viewed in color). Feature maps extracted by encoder are mapped into the word capsule vectors. Each group of action capsule then receives one of action labels that controls the collection of word capsule vectors and outputs representations encapsulating this action. A shared predictor updates action capsule vectors with different hidden states for each action and finally all vector representations are aggregated before sent to decoder to yield the final prediction.

**Capsule networks**: The concept of capsule was initially proposed in (Sabour et al. (2017)). Multiple layers of capsule units were employed to discover the hierarchical relationships of objects through routing algorithms so that CapsNet could output more robust features. Follow-up works (Kosiorek et al. (2019), Lenssen et al. (2018)) further enhanced capsule structure to better model invariant or equivariant properties. In this work, we argue that capsule networks also provide a ideal mechanism to map relationships back to pixel space and can be applied in a opposite way. As we already learn the part-whole relationships of action labels, we will show later that we can utilize capsule networks to control the generative behaviour of model.

## 3 OUR APPROACH: ACTION CONCEPT GROUNDING NETWORKS

We begin with defining the task of semantic action-conditional video prediction. Given an initial frame $x_0$ and a sequence of action labels $a_{1:T}$, the model is required to predict the corresponding future frames $x_{1:T}$. Each action label is a pre-defined semantic description of a spatiotemporal movement spanning over multiple frames in a scene like "*take the yellow cup on the table*". So technically, one can regard this task as an inverse problem of action recognition.

### 3.1 ACGN: MODULES

The ACGN model is composed of 4 modules including encoder $\mathcal{E}$, decoder $\mathcal{D}$, action capsule module $\mathcal{C}$ and recurrent predictor $\mathcal{P}$. The goal of our model is to learn the following mapping:

$$\hat{x}_{t+1} = \mathcal{D}(\mathcal{P}(\mathcal{C}(\mathcal{E}(x_t)|a_t)|h_t)) \tag{1}$$

where $x_t$, $a_t$ and $h_t$ are video frame, action labels and hidden states at time $t$. The overall architecture of our method is illustrated in Fig 2.

**Encoder**: At each timestep $t$, the encoder $\mathcal{E}$ receives visual input $x_t$ and extracts a set of multi-scale feature maps. We employ a convolutional neural network with an architecture similar to VGG16 (Simonyan & Zisserman (2014)). In order to encode sufficient spatial information of objects, the final down-sampling operations in VGG is replaced with resolution-preserving convolution layers.

**Action capsule module**: The action capsule module $\mathcal{C}$ is the core module of ACGN. The function of this module is to decompose each action label $a_t$ into words, to learn representations for each word from extracted feature maps, and to reorganize word capsules to represent semantic actions. There are two layers present: the word capsule layer and action capsule layer. After the feature maps are obtained from the image, they are fed into $\mathcal{K} \times \mathcal{N}$ convolutional filters, i.e. the word capsule layer, to

create $\mathcal{N}$ word capsule vectors of dimension $\mathcal{K}$. Here, $\mathcal{N}$ is the total number of words we pre-defined in the vocabulary or dictionary of action labels. Since verbs can be interpreted as spatiotemporal changes of relationships between objects, not only capsules for nouns but also capsules for verbs, like '*take*' or '*put*', are computed from the extracted feature maps.

The next layer in action capsule module is action capsule layer. Unlike the original CapsNet (Sabour et al. (2017)) which uses dynamic routing to determine the connections between two consecutive layers, connections between word and action capsule layer are directly controlled by action labels. We don't need to apply any routing-by-agreement algorithms as we already know the part-whole relationships in the case of words and actions. If we consider the following iterative procedure of dynamic routing between capsules $i$ and capsules $j$ of the next layer.

$$\mathbf{c}_j \leftarrow \text{softmax}(\mathbf{b}_j) \,, \quad \mathbf{v}_j \leftarrow \sigma(\sum_i c_{ij}\hat{\mathbf{u}}_{j|i}) \,, \quad b_{ij} \leftarrow b_{ij} + \hat{\mathbf{u}}_{j|i}\mathbf{v}_j \tag{2}$$

where $\mathbf{c}_j, c_{ij}$ are coupling coefficients, $\mathbf{v}_j$ is vector output of capsule $j$, $\hat{\mathbf{u}}_{j|i}$ is $j_{\text{th}}$ predictor vector computed linearly from capsule $i$ and $\sigma$ is the activation function. Our implementation is equivalent to set $\mathbf{c}_j$ as one-hot encoding vector determined by action labels and thus we no longer need to update $\mathbf{b}_j$ and $\mathbf{c}_j$ iteratively. More specifically, we decompose each action label into several meaningful clauses, represent each clause as one-hot encoding and establish connections through capsule-wise multiplication for each clause. Each clause will have its own dictionary recording all predefined words or concepts and one-hot encoding vectors are produced based on these dictionaries. Therefore, each action capsule outputs instantiation parameters of some clause in one of ongoing actions. It is worth noticing that ACGN is allowed to have multiple concurrent actions in a scene. In this case, we build additional groups of action capsules to represent other actions.

**Predictor**: The recurrent predictor $\mathcal{P}$, implemented as a stack of residual ConvLSTM layers, will then calculate the spatiotemporal evolution for each action label respectively. We concatenate outputs of action capsules from the same action label pixel-wisely before sending them to predictor. The memory mechanism of ConvLSTM is essential for ACGN to obtain the ability to remember its previous actions and to recover the occluded objects. To prevent interference between concurrent actions, hidden states are not shared between actions. The outputs of predictor for all action labels are added point-wisely along with pixel-wisely flatten word capsule vectors.

**Decoder**: Finally, the decoder $\mathcal{D}$ aggregates the updated latent representations produced by predictor and multi-scale feature maps from encoder to generate the prediction of next frame $\hat{x}_{t+1}$. The decoder is a mirrored version of the encoder with down-sampling operations replaced with spatial up-sampling and additional sigmoid output layer.

**Training**: We train our model by minimizing the average squared error the between the ground truth frames and the predictions. At the inference phase, the model will use its previous predictions as visual inputs instead except for the first pass. Hence, a training strategy called scheduled sampling (Bengio et al. (2015)) is adopted to alleviate the discrepancy between training and inference.

### 3.2 CONNECTION BETWEEN ACTION CAPSULE AND ACTION GRAPH

The proposed ACGN is related to a concurrent work AG2Vid (Anonymous (2021)) in which the authors provide the formalism of action-graph-based video synthesis. Action graph can be intuitively viewed as a temporal extension of scene graph, which is another appropriate structure to represent actions. In practice, nodes in both scene and action graphs correspond to object bounding boxes with their object categories and actions are represented as labeled directed edges between object nodes.

We would like to argue that the action capsule module also embeds a sparse graph structure into its output. To model the spatiotemporal transition of action graph, a graph convolution neural network (Kipf & Welling (2016)) that outputs bounding box layouts of next frame is employed in AG2Vid. If we treat word capsules as nodes in a graph, the routing mechanism controlled by action labels actually carries out a similar computation of $A \times H$ part of GCN where $A$ is adjacency matrix, $H$ are node features and $\times$ is matrix multiplication. This is because each row in the adjacency matrix of an action graph can be viewed as a concatenation of one-hot encoding vectors. In contrast to AG2Vid, our ACGN adopts three-way tensors that preserves the spatial structure to represent the entity and actions in the scene instead of using bounding boxes and category embeddings. Also spatiotemporal transformations in our method are mainly modeled by the following recurrent predictor.

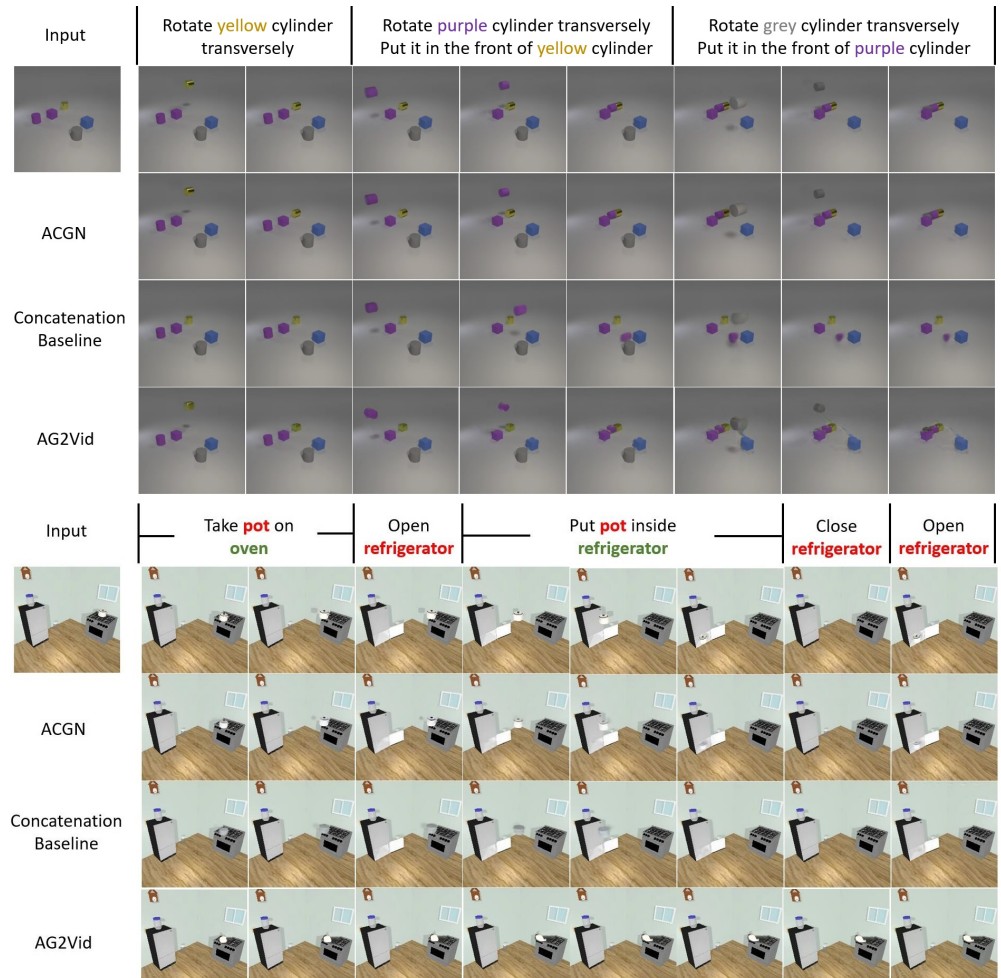

Figure 3: The qualitative comparison of all methods on CLEVR-Building-blocks and Sapien-Kitchen. The first row of each figure is the groundtruth sequence

## 4 DATASETS FOR ACTION CONDITIONAL VIDEO GENERATION

In this study, we create two new synthetic datesets, CLEVR-Building-blocks and Sapien-Kitchen, for evaluation instead of using existing datasets. This is because most video datasets either don't come with semantic action labels (Babaeizadeh et al. (2017)) or fail to provide necessary visual information in their first frames due to egomotions and occlusions (Hundt et al. (2018)), especially those inevitable occlusions brought by hands or by robotic arms. Although there are several appropriate datasets like Penn Action (Zhang et al. (2013)) and KTH (Schuldt et al. (2004)), they all adopt the same single-entity setting which actually indicates they can be solved by a much simpler model. To tackle the above issues, we design each video in our datatsets as a depiction of certain atomic action performed by an invisible agent with objects which are observable in the starting frame. Furthermore, we add functions to generate bounding boxes of all objects for both datasets in order to train AG2Vid.

### 4.1 CLEVR-BUILDING-BLOCKS DATASET

As its name would suggest, CLEVR-Building-blocks dataset is built upon CLEVR environment (Johnson et al. (2017)). For each video, the data generator first initializes the scene with 4 - 6 randomly positioned and visually different objects. There are totally 32 combinations of shapes, colors and materials of objects and at most one instance of each combination is allowed to appear in a video sequence. The agent can perform one of the following 8 actions on objects $\mathcal{O}_A$ and $\mathcal{O}_B$: *Pick $\mathcal{O}_A$, Pick and Rotate $\mathcal{O}_A$ transversely / longitudinally, Put $\mathcal{O}_A$ on $\mathcal{O}_B$, Put $\mathcal{O}_A$ on the left / right side of $\mathcal{O}_B$, Put $\mathcal{O}_A$ in the front of / behind $\mathcal{O}_B$.* Each training sample contains a video of three consecutive *Pick-* and *Put-* action pairs and a sequence of semantic action labels of every frame.

## 4.2 SAPIEN-KITCHEN DATASET

Compared with CLEVR-Building-blocks, Sapien-Kitchen Dataset describes a more complicated environment in the sense that: (a). It contains deformable actions like *"open"* and *"close"*; (b). The structures of different objects in the same category are highly diverse; (c). Objects can be initialized with randomly assigned relative positions like *"along the wall"* and *"on the dishwasher"*. We collect totally 21 types of small movable objects in 3 categories, *bottle, kettle* and *kitchen pot*, and 19 types of large openable appliances in another 3 categories, *oven, refrigerator* and *dishwasher*, from Sapien engine (Xiang et al. (2020)). The agent can perform one of the following 6 atomic actions on small object $\mathcal{O}_s$ and large appliance $\mathcal{O}_l$: *Take $\mathcal{O}_s$ on $\mathcal{O}_l$, Take $\mathcal{O}_s$ in $\mathcal{O}_l$, Put $\mathcal{O}_s$ on $\mathcal{O}_l$, Put $\mathcal{O}_s$ in $\mathcal{O}_l$, Open $\mathcal{O}_l$* and *Close $\mathcal{O}_l$*. Another 3 composite action sequences are defined as follows: *"Take_on–Put_on"*, *"Take_on–Open–Put_in–Close"*, *"Open–Take_in–Close"*.

## 5 EXPERIMENTAL EVALUATION

### 5.1 ACTION-CONDITONAL VIDEO PREDICTION

**Baselines and setup**: We evaluate the proposed model on CLEVR-Building-blocks and Sapien-Kitchen Datasets. AG2Vid (Anonymous (2021)) is re-implemented as the baseline model because it is the most related work. Unlike our method which only needs visual input and action sequence, AG2Vid also requires bounding boxes of all objects and progress meters of actions, i.e. clock edge, for training and testing. Furthermore, we conduct an ablation study by replacing action capsule module with the concatenation of features and tiled action vector, which is commonly used in low-level action-conditional video prediction (Finn et al. (2016)), to show the effectiveness of our module. To make fair comparison, the number of parameters of ACGN and its concatenation-based variant are the same.

**Metrics**: To estimate the fidelity of action-conditional video prediction, MSE, SSIM (Wang et al. (2004)) and LPIPS (Zhang et al. (2018b)) are calculated between the predictions and groundtruths. We also perform a human study to assess the accuracy of performing the correct action in generated videos for each model. The human judges annotate whether the model can identify the desired objects, perform actions specified by action labels and maintain the consistent visual appearances of all objects in its generations and only videos meeting all three criterions are scored as correct.

**Results**: The quantitative comparisons of all methods are summarized in Table 1. The ACGN achieves the best scores on all metrics without access to additional information like bounding boxes, showing the superior performance of our action capsule module. The qualitative analysis in Fig 3 further reveals the drawbacks of other baselines. For CLEVR-Building-blocks, the concatenation-based variant fails to recognize the right objects due to its limited inductive bias. Although AG2Vid has no difficulty in identifying the desired objects, assumptions made by flow warping are too strong to handle rotation and occlusion. Consequently, the adversarial loss enforces AG2Vid to fix these errors by converting them to wrong poses or colors. These limitations of AG2Vid will be further amplified in a more complicated environment, i.e. Sapien-Kitchen. The same architecture used for CLEVR can only learn to remove the moving objects from their starting positions in Sapien-Kitchen because rotation and occlusion occur more often. The concatenation baseline performs better by showing correct generation of open and close actions on large appliance. Yet, it still fails to produce long-term consistent predictions as the visual appearances of moving objects are altered. On the contrary, ACGN can authentically depict the correct actions specified by action labels on both datasets.

### 5.2 COUNTERFACTUAL GENERATION

**Counterfactual generation**: The most intriguing application of our ACGN is counterfactual generation. More specifically, counterfactual generation means that our model will observe the same starting frame but receive different valid action labels to generate the corresponding future frames.

**Results**: The visual results of counterfactual generations on each dataset are displayed in Fig 4. As we can see, our model successfully identifies the desired objects, plans correct trajectories toward the target places and generates high-quality imaginations of counterfactual futures. It is also worth noticing that all displayed generations are long-term generations , i.e. more than 30 frames are predicted for each sequence. Our recurrent predictor plays an very important role in sustaining the spatiotemporal consistency and in reconstructing the fully-occluded objects.

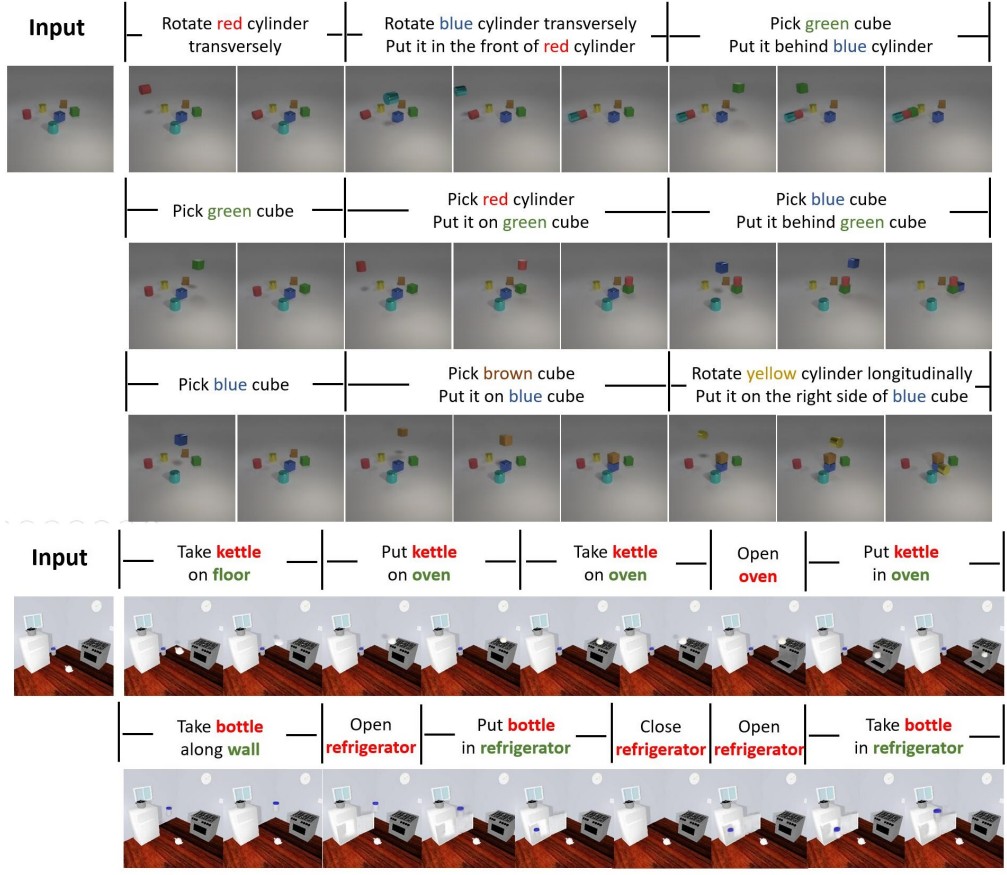

Figure 4: Counterfactual generations on CLEVR-Building-blocks and Sapien-Kitchen. Descriptions above images indicate the action labels we input. All images are predictions except for input frames.

| Model | CLEVR-Building-blocks | | | | Sapien-Kitchen | | | |
|---|---|---|---|---|---|---|---|---|
| | SSIM↑ | MSE↓ | LPIPS↓ | Accuracy↑ | SSIM↑ | MSE↓ | LPIPS↓ | Accuracy↑ |
| Copy-First-Frame | 0.962 | 251.38 | 0.1320 | - | 0.951 | 152.87 | 0.0393 | - |
| Concatenation Baseline | 0.961 | 226.53 | 0.1301 | 50.8% | 0.962 | 23.13 | 0.0232 | 52.4% |
| AG2Vid  Anonymous (2021) | 0.956 | 58.67 | 0.0399 | 78.8% | 0.947 | 270.87 | 0.0684 | 5.2% |
| ACGN | **0.983** | **43.52** | **0.0303** | **95.2%** | **0.971** | **11.16** | **0.0178** | **86.4%** |

Table 1: Quantitative evaluation of all methods on CLEVR-Building-blocks and Sapien-Kitchen. All metrics are averaged frame-wisely except for accuracy.

### 5.3 CONCURRENT ACTIONS, ADAPTATION AND DETECTION

We further explore other interesting features of our ACGN on Sapien-Kitchen dataset. We first demonstrate that our ACGN is capable of generating video sequences depicting concurrent actions, which can be considered as out-of-distribution generations because our model only observes single-action videos during the training. We also try to evaluate how quickly our model can be adapted to new objects. It turns out for each new object, our trained ACGN only requires a few training video examples to generate decent results. Finally, to verify that our model encodes the spatial information, we add SSD (Liu et al. (2016)) head after the frozen encoder to conduct object detection.

**Concurrent actions**: Concurrent actions means multiple action inputs at the same time. Generating concurrent-action videos needs to employ copied action capsules and parallel hidden states. As illustrated in Fig 5, our ACGN can linearly integrate the action information in the latent space and correctly portray two concurrent actions in the same scene.

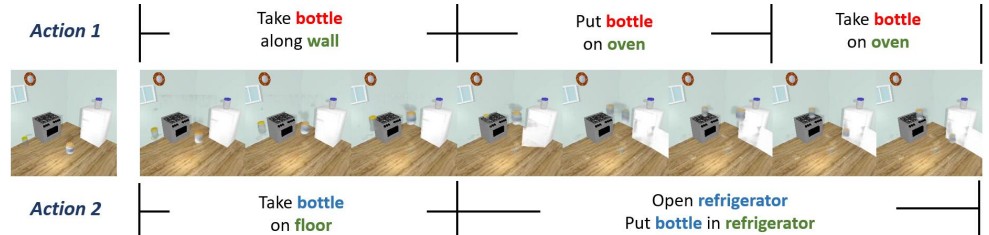

Figure 5: Concurrent-action video generations on Sapien-Kitchen datasets. Descriptions above and below images indicate two concurrent action labels we input into our ACGN.

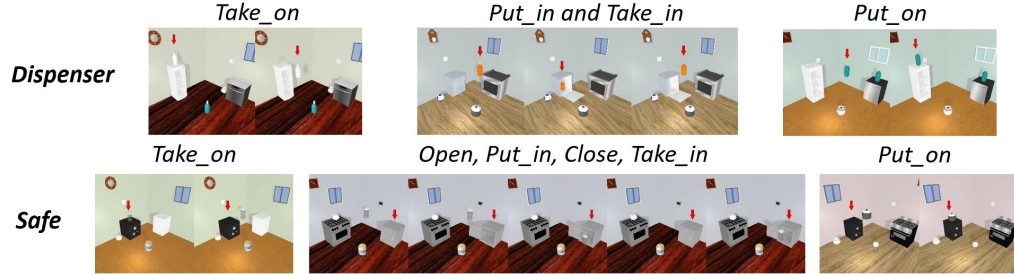

Figure 6: Generations on new objects, *dispenser* and *safe*. Red arrows point to new objects in images

| Method | Oven | Refrigerator | Dishwasher | Bottle | Kettle | Kitchen pot | mAP |
|---|---|---|---|---|---|---|---|
| ACGN encoder + SSD | 92.75 | 94.56 | 90.89 | 83.25 | 77.18 | 81.32 | 86.66 |

Table 2: Quantitative measures of object detection on Sapien-Kitchen in terms of average precision.

**Adaptation**: We add a new openable category "*safe*" and a new movable category "*dispenser*" into Sapien-Kitchen and generate 100 video sequences for each new object showing its interaction with other objects. Approximately, there are about 5 new sequences created for each new action pair between 2 objects. Blank word and action capsule units for new categories are attached to trained ACGN and we finetune it on this small new training set. Fig 6 shows that even with a few training samples, ACGN is accurately adapted to new objects and produces reasonable visual evolution. This is because, with the help of action capsules, ACGN can disentangle actions into relatively independent grounded concepts . When it is learning new concepts, ACGN can reuse and integrate the prior knowledge learnt from different scenarios.

**Object detection**: The quantitative results of object detection is provided in Table 2 and visualization can be found in Fig 7 in Appendix. We observe that the features learnt by ACGN can be easily transferred for detection as our video prediction task is highly location-dependent. This result indicates that utilizing bounding boxes might be a little redundant for some video tasks because videos already provide rich motion information that can be used for salient object detection.

## 6 CONCLUSION

In this work, we propose the new task of semantic action-conditional video prediction and introduce two new datasets that are meant to bridge the gap towards a robust solution to this task in complex interactive scenarios. ACGN, a novel video prediction model, was also designed by utilizing the idea of capsule network to ground action concept for video generation. Our proposed model can generate alternative futures without requiring additional auxiliary data such as bounding boxes, and is shown to be both quickly extendible and adaptable to novel scenarios and entities. It is our hope that our contributions will advance progress and understanding within this new task space, and that a model robust enough for real-world applications (i.e. in robotic systems) in perception and control will be eventually proposed as a descendant of this work.

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

# A    VISUALIZATION OF OBJECT DETECTION

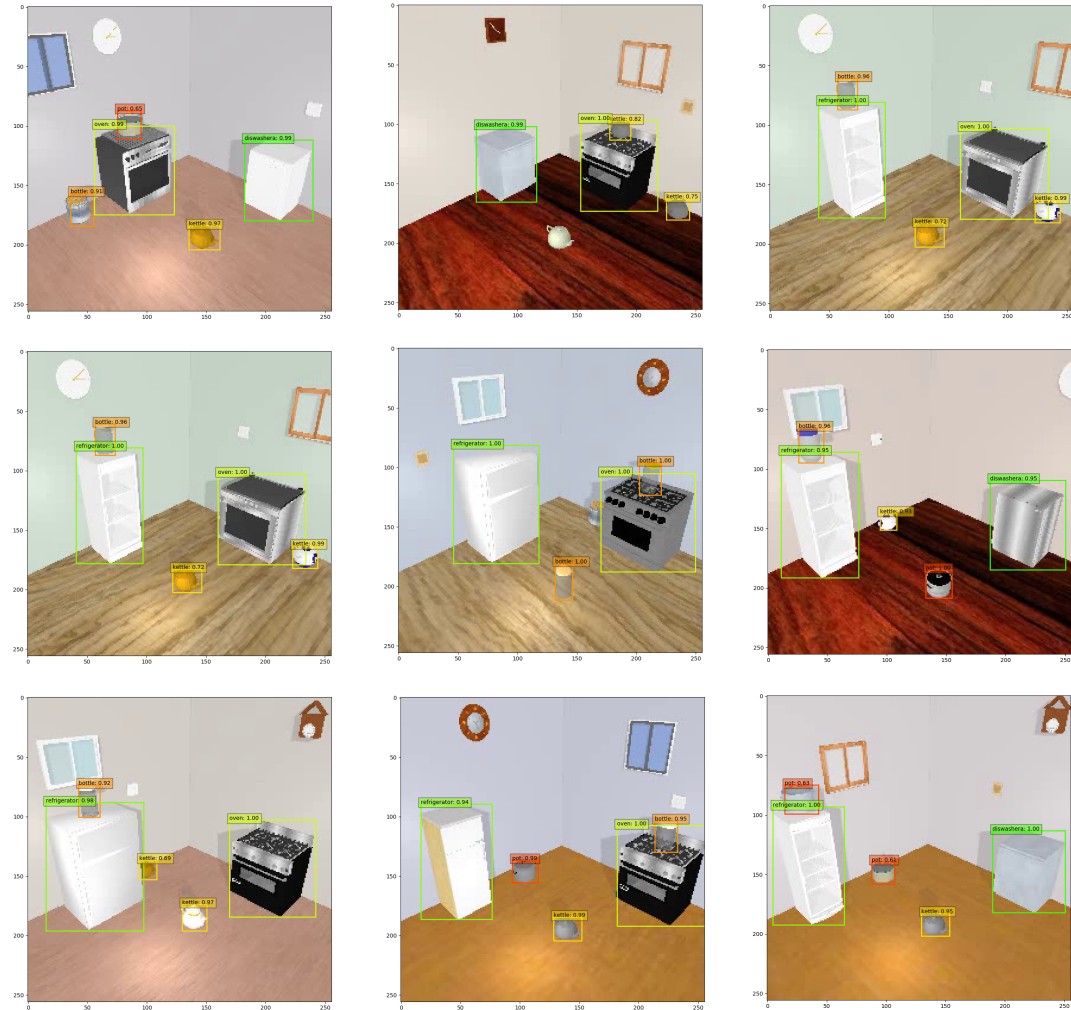

Figure 7: Visualization of 2D Object Detection on Sapien-Kitchen.

## B    MORE ACTION-CONDITIONAL VIDEO PREDICTIONS OF ACGN ON CLEVR-BUILDING-BLOCKS DATASET

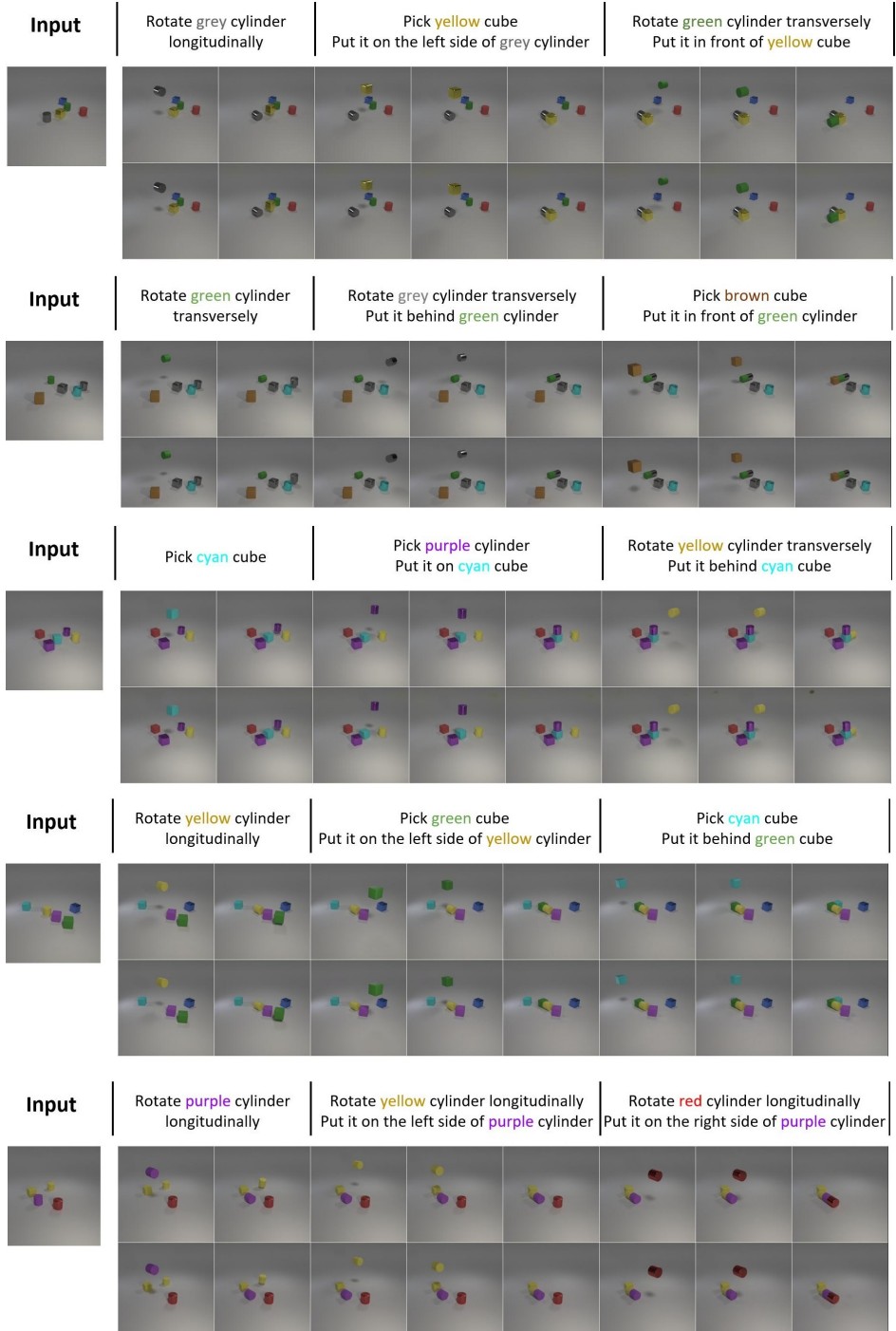

Figure 8: Action-conditional video prediction of ACGN on CLEVR-Building-blocks dataset.

## C    MORE ACTION-CONDITIONAL VIDEO PREDICTIONS OF ACGN ON SAPIEN-KITCHEN DATASET

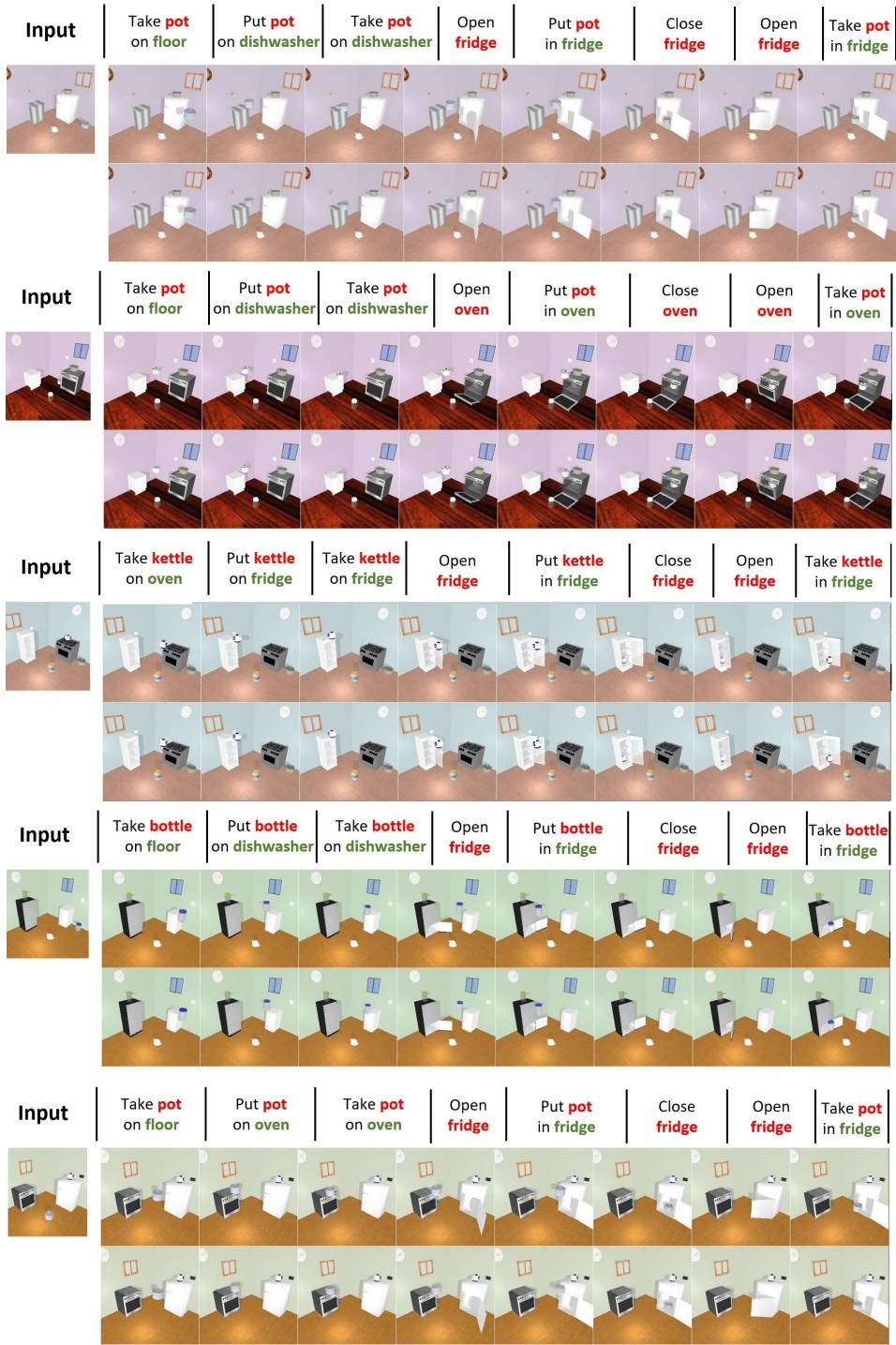

Figure 9: Action-conditional video prediction of ACGN on Sapien-Kitchen dataset.

