# OpenReview forum: "Action Concept Grounding Network for Semantically-Consistent Video Generation"
_ICLR.cc/2021/Conference — Reject_

### Official Review · AnonReviewer1 · 2020-10-27

**Rating:** 5
**Confidence:** 5

**Review:**

Summary


Pros
+ New interesting task for video synthesis
+ New capsule based neural network to learn relationships
+ Outperforms baselines in pixel based metrics and user study

Comments / Questions:
- Stopping criteria:
I may have missed it, but I cannot find what is the stopping criteria of the video generation. Is it when the generated frame stops changing? If not, what happens if you let the network generate frames infinitely, does the frame become static once the task is accomplished?

- Number of parameters:
The proposed method needs additional parameters for each of the capsule blocks feature computation. Does the concatenation match the number of parameters of the proposed method? Or is everything kept the same except for the concatenation replacing the action capsule?

- Difference between the testing and training data:
It’s obvious that for the input instructions, the network must be able to see the before to generalize, however, the object arrangement is a different story. How much does the training and testing data differ in object spatial arrangement? Is it safe to assume that the same exact test image won’t be in the training set?

- Capsule module embeds sparse graph structure:
The authors mention that they argue about sparse graph structure happening in their network. Is there a way to visualize the internal state of the network to verify this?


Conclusion:
This paper presents an interesting task that has the potential of sparking research in this direction. In addition, the authors propose an interesting architecture that generates extremely good video given some simple instructions. There are a few questions I have listed above, however, the paper to me seems interesting enough consider for acceptance.

###########################
 POST REBUTTAL DECISION
###########################

After reading other reviews and the rebuttal, I have concerns related to the stochastic trajectories mentioned by R2. The fact that the authors confirmed that all semantic actions have the same number of steps makes me question potential overfitting. I would imagine that some actions should take less steps than others based on the objects that are being interacted with, and potential multiple trajectories when performing the same semantic action. I still think the task is interesting, but the setup seems not appropriate to claim a general concept from the current method. Therefore, I have decided to lower my score.

---

> ### Author Response · Authors · 2020-11-18
> **Response to Reviewer #1**
>
> We would like to thank R1 for your encouraging and constructive comments.
>
> **Stopping criteria**: The generative process will stop when there is no new action input received. Also, by design, the same actions in different scenarios will last the same number of frames so that the model can know when to finish actions. We are allowed to do so because action-conditional video prediction models are usually used for visual planning, not for dynamics, and what we are interested in are goal frames and intermediate positions of moving objects.
>
> **The number of parameters**: Yes. The number of parameters for ACGN and its concatenation variant in ablation study are the same.
>
> **Dataset**: Please refer to the comprehensive comment we post about datasets first.
> https://openreview.net/forum?id=4_57x7xhymn&noteId=uY_jZdhWFFS
> We shared the same concern about possible overlapping between training and testing data. To make sure that such overlapping would not happen, we recorded the layouts of all video sequences and compared the layouts between training and testing. It turned out that none of the testing sequences had appeared in training data.
>
> **Capsule module embeds sparse graph structure**: The graph structures used in GCN or GNN are represented by adjacency matrix and finding the **adjacency matrix** is sufficient to verify the existence of graph structures. In the action capsule module, we multiply word capsules with one-hot encoding vectors of clauses for routing, which can be viewed as another form of adjacency matrix of action graph. This is because each row of the adjacency matrix of the action graph is literally a concatenation of one-hot encoding vectors.

---

### Official Review · AnonReviewer3 · 2020-10-28

**Rating:** 5
**Confidence:** 4

**Review:**

###Summary###

The paper proposes the video prediction model that can handle multiple objects.
The proposed model is built from attention-based capsule network, and it generates video in a hierarchical feature computation.
Specifically, it first computes features for each concept(objects, actions), and then generate features for future image sequence corresponding to given action labels.



###Pros###

Based on the capsule network, the model learns to discern each concept without any need for labels such as bounding boxes.



###Questions###

-
The explanation of the method is insufficient.
Is the length of the action label sequence(a1:T) same with the length of the future frame sequence(x1:T)?
Could you explain more about the one-hot encoding vector? Is code for bottle and oven same? (Figure 2)
It would help in understanding if more equations are specified.
Specifying the dimension of each feature in the figure 2 also might be helpful too.

-
Since the datasets used are relatively simple, it is possible that the trained model was overfit.
As it can be seen in the figure 3, prediction results from ACGN is quite similar to the ground truth sequence.
Could you represent some results which prove that model has not just memorized the dataset?


###Minor comments###

-
There are some typos

(1) 8th line in abstract : AGCN -> ACGN

(2) title of 3.1 section : AGCN -> ACGN

(3) 3rd line in Results paragraph of 5.1 section : quantitative -> qualitative

---

> ### Author Response · Authors · 2020-11-18
> **Response to Reviewer #3**
>
> Thank R3 for your helpful comments.
>
> **Method**: We will explain more about our method by giving a specific example in *Sapien-Kitchen*. Say we have a 16-frame video sequence performing two actions *“take pot on dishwasher”* and then *“put pot on oven”*. The first action will last the 5 frames after the initial frame and the second will last the next 10 frames.(1+5+10=16) In this case, action labels $a_{1:5}$ are the same, which is *“take pot on dishwasher”*, and similar for $a_{6:15}$. We will input the corresponding action label at each timestep. So, the length of action label sequences is the same as the number of future frames. Then we will translate each action label into one-hot vector embedding. We decompose each action into several clauses. For example, *“take pot on dishwasher”* will be split into *‘take_on’,’pot’* and *‘dishwasher’*. And for each clause, it will find its one-hot encoding vector according to the predefined dictionary. For verbs, the predefined dictionary is *{‘take_on’,’put_on’,’open’,‘close’,‘take_in’,’put_in’}*. Thus, the one-hot vector for *‘take_on’* is *‘100000’*. We admit that the code in Figure 2 is a little confusing but we deliberately drew so that readers can realize words from different clauses don’t share the same dictionary. We have added several sentences in the paper to give more clear explanation of how to produce one-hot encoding vectors.
>
> **Datasets**: Please refer to the comprehensive comment we posted about datasets first.
> https://openreview.net/forum?id=4_57x7xhymn&noteId=uY_jZdhWFFS
> We think our calculation proves that our datasets are not simple and it is impossible to have the overfitting or memorization issue. The results look good because our model is very powerful. If this was caused by memorization, then the concatenation-based variant would also overfit and give us good predictions. You can find more visualization results in appendix and https://iclr-acgn.github.io/ACGN/ where some predictions are flawed.
>
> We fixed all typos mentioned by R3 in the paper. Finally, we would be curious to hear more R3’s thoughts on other aspects our paper. Thank you.

---

### Official Review · AnonReviewer2 · 2020-10-30
**An interesting paper with limited technical contributions**

**Rating:** 5
**Confidence:** 5

**Review:**

This paper proposes a new task of semantic action-conditioned video prediction. The purpose of this task is to generate future frames given objects and action pairs. The paper presents a new model for this task, which makes use of the capsule networks to learn hierarchical relationships between objects. The object-action relations are represented with three-way tensors. In general, I think it is an interesting paper. But I still have some concerns about the significance of the new task, the novelty of the technical contributions, and the experiments as well.
1. The proposed task emphasizes the use of semantic actions instead of low-level actions in previous video prediction work. As we have seen the success of low-level action-conditioned video prediction in the field of visual planning [Ebert et al., 2017], how significant is the proposed task in practical applications?
[Ebert et al., 2017] Self-Supervised Visual Planning with Temporal Skip Connections.
2. The proposed model is mainly based on the previous Capsule networks. What is the most challenging part of the new task? In addition to using Capsule networks to capture three-way object-action relations, what is the key insight of the proposed model?
3. Different from the low-level action-conditioned video prediction task, the action labels are ambiguous sometimes. As shown by the case in Figure 2, the action “put pot on oven” may have multiple plausible moving trajectories over several timestamps, and may finally lead to many plausible positions of the pot on the oven. All these future trajectories and final positions are reasonable since they reflect the uncertainty of the action labels. The question is how does the proposed model cope with this uncertainty? Can the model produce diverse future frames?
4. How does the model perform under more action labels and object labels? On the kitchen data set, the authors used only six categories of objects and six categories of actions, which means that the model only needs to consider 36 types of input actions. This may not be a very challenging problem in my view.
5. It would be nice to have more discussion about the generalization ability of the model. Can the model be generalized to action descriptions with more than two objects?

---

> ### Author Response · Authors · 2020-11-18
> **Response to Reviewer #2**
>
> Thank R2 for your constructive feedbacks.
>
> **Q1**: Thank R2 for referencing this impressive paper. We have added this paper to our citation as it is highly related to our paper. We will clarify the significance of our new task by showing its difference from its low-level counterpart in practical applications. Intuitively, **low-level and semantic action-conditional video prediction models are tackling two different aspects of MPC problem**, i.e. low-level action-conditional video prediction simulates the interaction process to further plan toward the **given goal pixels**, while the semantic counterpart,  given semantic instructions, can tell where the **goal** and intermediate positions are. More specifically, in the paper of SNA [1], users need to specify the starting, goal and perhaps intermediate pixels in the image to tell the model what to move and where the goal is. Although SNA is successful in visual planning, it still requires humans to help locate the object and goal area because it doesn’t fully understand the scene.
>
> Our semantic task, on the other hand, requires the model to infer the locations of the desired object and its goal area given more abstract action instructions.  It also involves multi-entity relationship reasoning because the model needs to understand the actions containing relative positions like *“take bottle on the dishwasher”*. This means our task requires the model to have a higher-level understanding of scene and long-term planning. In general, the relationship between low-level and semantic task is more like complementary. A combination of low-level and semantic action-conditional video prediction models is also possible in which the semantic part provides intermediate position and goal pixels and the low-level part simulates the local transformation.
>
> **Q2**: The main challenge of this new task, as mentioned in the third paragraph of the introduction section, is **how to inform the model of high-level semantic action information or how to ground action concepts**. We will explain more about why it is very difficult in the multi-entity settings. Essentially, the only self-supervisory signals we have in the task of video prediction are pixel changes between consecutive frames. However, if we take the action *“put bottle on the oven”* as an example, since the oven is not moving in this case, i.e. no pixel changes for the oven in most frames,  it will be extremely hard to make the model learn what and where the oven is. The concatenation-based method commonly used in low-level action-conditional video prediction will fail because it is designed to only encode the movement information of a single entity such as a robotic arm. Another way, as described in the concurrent work AG2Vid [2], is directly telling the model the location of each object with help of a pre-trained detector or ground-truth bounding box. But it actually simplifies the task and bounding boxes are insufficient to describe rigid body movement in 3D space.
>
> In this paper, we first argue that concatenation-based methods are not powerful enough to handle our newly designed task and using additional information such as bounding boxes are unnecessary if we find the proper architecture, i.e. action capsule, to condition the model with abstract action concepts. We demonstrate that capsules can be applied in a opposite way as a new approach to bridge language and vision. Another key insight of the proposed model is that action capsules perform much better than other baselines **because it can disentangle actions into relatively independent grounded concepts. As a result, ACGN can reuse and integrate the knowledge learnt from different scenarios.** Take the same *“put bottle on the oven”* as an example, although it is hard to learn where the oven is in this case, ACGN can take advantage of other cases where the oven is moving, e.g.*”open the oven”*, to help understand the concept of oven. This concept grounding property also gives ACGN very impressive generalization power.
>
>
> [1]. [Ebert et al., 2017] Self-Supervised Visual Planning with Temporal Skip Connections.
>
> [2]. [Anonymous] Compositional Video Synthesis with Action Graphs. https://openreview.net/forum?id=tyd9yxioXgO
>
> [3]. [Emily Denton and Rob Fergus, 2018] Stochastic video generation with a learned prior.
>
> [4]. [A. X. Lee et al., 2018] Stochastic adversarial video prediction.

---

> > ### Author Response · Authors · 2020-11-18
> > **Response to Reviewer #2 Part 2**
> >
> > **Q3**: This is a very good question. The current version of datasets only described deterministic trajectories of actions because we wanted to keep the same settings as the only available baseline AG2Vid [2] did. However, we did attempt to study the randomness issue before our submission. More specifically, we created a stochastic version of *Sapien-Kitchen* in which we added small random variations in trajectories and ending positions and trained the ACGN on it. The generative results are shown in the following link.
> > https://iclr-acgn.github.io/ACGN/data/random_traj.gif
> > where the left column is ground truth and right column is prediction. As we can see, the moving objects will become a little more blurred because the model averages the possible futures when it predicts. Note that the key contribution of ACGN is its mechanism to map semantic actions to video generations, which means empirically it should remain compatible with other variational methods used for stochastic video generation. We tried to combine those modules proposed in SVG [3] and SAVP [4] with our ACGN, but the results were shown much worse than the original ACGN. https://iclr-acgn.github.io/ACGN/data/random_traj_svg.gif This indicates that modules from SVG may only work in short-horizon predictions and learning how to generate long-term random walks toward goal requires a different mechanism. In summary, we have evaluated the proposed method on the stochastic variant of *Sapien-Kitchen* and it could produce meaningful but a little more blurred predictions. However, it was unable to generate diverse trajectories and adding this function will be left for future work.
> >
> > **Q4**: Please refer to the comprehensive comment we posted about datasets first.
> > https://openreview.net/forum?id=4_57x7xhymn&noteId=uY_jZdhWFFS
> > The number of all **atomic actions** defined in *Sapien-Kitchen* is 48 because it involves multi-object actions like *“put A on B”*. **It should be made clear that the complexity of actions also comes from the combinations of atomic actions** since *“take A on B and put A on C”* is a different case from *“take A on B and put A inside C”*. For any video in Sapien-Kitchen, if we only consider the first *“take-put”* action pair in the categorical level, the number of possible combinations is $3^3\times3\times2\times3\times4=1944$. Thus, our dataset indeed describes a very complicated action space and our new task is very challenging for reasons we described above. As for more categories and labels, we tried to add two more categories for few-shot learning in section 5.3 and demonstrated that our model could be quickly adapted to new object categories with a few training samples. The generative results of concurrent actions could also be viewed as adding new action labels.
> >
> > **Q5**: The ACGN was shown to possess great generalization power because it could disentangle actions into relatively independent grounded concepts with the help of action capsules. As a result, it could extrapolate to unobserved scenarios such as concurrent actions and reuse its learnt prior knowledge to fast adapt to new cases. The **atomic actions** with more than two objects are not supported by the current version of ACGN because such language structures are not defined in the action capsules. However, **composite actions** like *“take A on B and put A on C”* can be regarded as actions with 3 objects. Also, we agree that it would be interesting to create an extension of the *Sapien-Kitchen* environment to have such problem instances coupled with atomic action descriptions involving more than two objects.
> >
> >
> > Finally, we want to emphasize that **the novelty of our work should be evaluated based on the prior arts**, especially AG2Vid. In this paper, we proposed a much harder and more abstract task that requires higher-level understanding of scene through self-supervised learning and two new datasets that are combinatorial complex. The ACGN outperformed AG2Vid significantly even in unfair comparisons, where AG2Vid requires access to ground truth bounding boxes but ACGN only needs simpler action instructions.  The hierarchical representation of semantic actions enabled by action capsules allowed ACGN to reuse and integrate the prior knowledge to understand concepts and to possesse great generalization power. We add the link to discussions of AG2Vid as below and recommend R2 to read it. https://openreview.net/forum?id=tyd9yxioXgO
> >
> > [1]. [Ebert et al., 2017] Self-Supervised Visual Planning with Temporal Skip Connections.
> >
> > [2]. [Anonymous] Compositional Video Synthesis with Action Graphs. https://openreview.net/forum?id=tyd9yxioXgO
> >
> > [3]. [Emily Denton and Rob Fergus, 2018] Stochastic video generation with a learned prior.
> >
> > [4]. [A. X. Lee et al., 2018] Stochastic adversarial video prediction.

---

### Public Comment · ~Yingxiao_Ye1 · 2020-11-16
**Code release for datasets**

Hi there, I think this is a great paper and the new task and datasets proposed in the paper are interesting. I wonder when you will release these two datasets so that I can play with them. I am looking forward to your reply.

---

> ### Author Response · Authors · 2020-11-18
> **Thank you for your interest.**
>
> Thank you for your interest in our work. We will release the data generation code soon. Please check the project page we posted in the abstract from time to time.

---

### Author Response · Authors · 2020-11-18
**Concerns about complexity of datasets**

We want to thank all reviewers for their efforts and time in reviewing our paper.

We find that all reviewers have concerns about complexity of the proposed datasets. Hence, we decided to write a comprehensive comment to explain why our datasets are actually very complicated. In short, **the complexity of the proposed datasets primarily comes from combinatorial explosion**. We start with calculating the complexity of layouts of objects in the starting frame for each dataset.

For *CLEVR-Building-Blocks*, we have 32 different types of objects and 4-6 of them will be initialized in the scene. It is easy to calculate the number of possible combinations of *N* objects present in the scene out of 32. For *N*=6, the number of possible combinations is $32!/(26!\times6!)= 901692$. Note that we have not considered any continuous initialization parameters such as random positions, lighting and camera poses.

For *Sapien-Kitchen*, we initialized each scene with 2 large appliances along the wall and 3 small objects randomly located in the scene. The calculation for the number of combinations is a little more complicated, which is $21^3\times(7\times12+12\times13)=2222640$. Similarly, continuous initialization parameters including background, lighting and initial positions are not considered.

In practice, we generated 20k video sequences for *CLEVR-Building-Blocks* and 30k for *Sapien-Kitchen* for training. Even if we only consider the complexity of layouts of objects in the starting frame for now. Our generated datasets are just not large enough to cover all possible cases. Thus, overfitting or memorization will not happen in our datasets. Otherwise, concatenation-based methods would memorize or overfit as well but in fact it performed much worse than ACGN.

Note that we have not considered the combination of possible action instruction sequences yet. Consider *CLEVR-Building-Blocks* as an illustration, if we have 6 objects presented in the scene, the number of possible actions for first take-put action pair is $6\times5\times5=150$. The number of second and third action pair will be at least $4\times5$ and $3\times5$, dependent on the shapes of the first two objects. Thus, the number of possible action sequences is at least 45,000.

Furthermore, we want to emphasize that **in this paper we only focus on the multi-entity settings which are more difficult and more important for visual planning and many real-world video datasets with action descriptions are not qualified because they adopt single-entity settings**. For example, Something Something dataset used in AG2Vid only describes movements of single-entity and in fact it can be easily solved by existing low-level action-conditional video prediction methods.

To sum up, given the combinatorics, our datasets are in fact more complicated than the other comparable environments used in reinforcement learning.

---

### Decision · Program_Chairs · 2021-01-07
**Final Decision**

**Decision:**

Reject

**Comment:**

This paper presents work on semantic action-conditioned video prediction.  The reviewers appreciated the interesting task and use of capsule networks to address it.  Concerns were raised over generalization ability of the proposed approach, points on clarity, scalability, and handling of uncertainty/diversity by the method.  After reading the authors' response, the reviewers engaged in discussion.  Over the course of this discussion, the reviewers converged on a reject rating, noting that the concerns raised above were not sufficiently addressed to warrant publication at this stage.